# Highly Parallel Deep Ensemble Learning

## Abstract

In this paper, we propose a novel highly parallel deep ensemble learning, which leads to highly compact and parallel deep neural networks. The main idea is to first represent the data in tensor form, apply a linear transform along certain dimension and split the transformed data into different independent spectral data sets; then the matrix product in conventional neural networks is replaced by tensor product, which in effect imposes certain transformed-induced structure on the original weight matrices, e.g., a block-circulant structure. The key feature of the proposed spectral tensor network is that it consists of parallel branches with each branch being an independent neural network trained using one spectral subset of the training data. Besides, the joint data/model parallel amiable for GPU implementation. The outputs of the parallel branches, which are trained on different independent spectral, are combined for ensemble learning to produce an overall network with substantially stronger generalization capability than that of those parallel branches. Moreover, benefiting from the reducing size of inputs, the proposed spectral tensor network exhibits an inherent network compression, and as a result, reduction in computation complexity, which leads to the acceleration of training process. The high parallelism from the massive independent operations of the parallel spectral subnetworks enable a further acceleration in training and inference process. We evaluate the proposed spectral tensor networks on the MNIST, CIFAR-10 and ImageNet data sets, to highlight that they simultaneously achieve network compression, reduction in computation and parallel speedup.

## 1 Introduction

Deep neural networks (DNNs) [1] have made impressive successes in many applications, such as computer vision [2][3][4], online game [5][6][7], natural language processing [8][9][10], autonomous driving [11][12][13], and robotics [14][15][16]. However, DNNs are memory-intensive and computation-intensive, which are two major challenges for wider adoption, e.g., in Internet of Things (IoT) applications [17]. Modern DNNs may have billions of parameters that consume excessive amount of memory and usually require long training time. For example, AlexNet [18] consists of three fully-connected layers and five convolutional layers, containing 60 million parameters and consuming about 250 MB of memory and about 40 hours for training on ImageNet data set [19].

To mitigate the impact of memory and computation intensive, there are two types of techniques most investigated by researchers, namely model compression and parallel training. The model compression methods, which represents the conventional neural layers in DNNs by compact layers, such as the strucred linear layers, tensor layers, *etc*., are one of the most efficient methods to reduce memory consumption of DNNs. Besides, the proposed parallel training methods can be generally categorized into data parallelism and model parallelism methods. It should be noted that tensor layer provides potential parallelism for efficient GPU computing as well as the reduction of the

Table 1: Different tensor layers. An $N^2 \times N^2$ weight matrix is organized into a $d$-th order tensor with the size of each dimension $n$, i.e., $N^4 = n^d$, and $r$ is the tensor rank.

| Methods | Model size | Inference time |
|---|---|---|
| Fully connected layer | $O(N^4)$ | $O(N^4)$ |
| Low-rank matrix [29] | $O(N^2 r)$ | $O(N^2 r)$ |
| CP tensor layer [31] | $O(nr)$ | $O(nr)$ |
| Tucker tensor layer [32] | $O(nr + r^3)$ | $O(nr)$ |
| HT tensor layer [32] | $O(dn^2 r + r^d)$ | $O(dnr^2 N^2)$ |
| Tensor-train layer [29] | $O(dnr^2)$ | $O(dr^3 N^2)$ |

memory consumption. Thus, it naturally motivates us to employ tensor layer to achieve both the model compression and parallel computing.

In this paper, we propose a unified approach that simultaneously achieves both model compression and parallel learning *without* communication overhead. The key technique is a novel spectral tensor layer that enables a joint *data/model-parallel* implementation of a DNN as follows: 1) The training data set is split into multiple orthogonal spectral sets; 2) The neural network is split into parallel branches with each branch being a conventional neural network, that are trained asynchronously and independently on the corresponding spectral sets; 3) The outputs of the parallel branches are finally combined to yield an overall neural network with substantially stronger generalization capability than that of those parallel branches.

The remainder of this paper is organized as follows. Section 2 presents an brief overview of the study on the model compression and parallel computing. Section 3 presents multiple spectral tensor layers, including fully connected layers, convolution layers and recurrent layers. Section 4 presents the experimental results and we conclude this paper in Section 5.

## 2 Related Works

**Network compression:** Consider a linear layer that is a central building block of modern DNNs, where an input $x \in \mathbb{R}^{N^2}$ (e.g., an $N \times N$ image) is transformed by a weight matrix $W \in \mathbb{R}^{N^2 \times N^2}$ [1], i.e., $y = Wx \in \mathbb{R}^{N^2}$. The memory size for storing W is $O(N^4)$ and the computational complexity of the matrix-vector product is also $O(N^4)$, both are very demanding for smartphones, robots, and embedded devices.

Many works [20][21][22][23] showed that over $95\%$ of the parameters are redundant, thus the network can be greatly compressed. The *structured linear layer* imposes certain structures on $W$, including circulant [21][22][24], circulant-block [25][26], and Teoplitz-like structures [21][27][23]. For example, for a circulant weight matrix $W$ [21][22][27][24], the memory size becomes $O(N^2)$ and the computational complexity becomes $O(N^2 \log N)$, at the expense of a slight drop in the inference performance.

On the other hand, the *tensor layer* exploits different types of low-rank tensor representations of the weight matrix $W$ [28][29][30][31][32]. For example, if the weight matrix $W \in \mathbb{R}^{N^2 \times N^2}$ has rank $r \ll N^2$, such that $W = AB$, $A \in \mathbb{R}^{N^2 \times r}$, $B \in \mathbb{R}^{r \times N^2}$, then the memory size becomes $O(2N^2 r)$, and the computational complexity becomes $O(2N^2 r)$. Table 1 summarizes the model size and computational complexity for various tensor representation schemes.

**Parallel training**: Existing works on parallel machine learning take advantage of either data-parallelism [33][34][35] or model-parallelism [36][37][38] or both [39]. In the data-parallel approach, the data are distributed among multiple processors that apply the same model [33]. The processors periodically exchange the outputs and gradients. In the model-parallel approach, exact copies of the whole data set are processed by multiple processors that operate in parallel on different parts of the same model [36]. There are frequent aggregation of model parts and distribution of gradients [40].

From the above discussion, we see that the computation associated with a DNN can be speed up via two separate ways: one is through model compression so that reduced number of weight parameters leads to reduced number of multiplications and additions; and the other is through parallel computing,

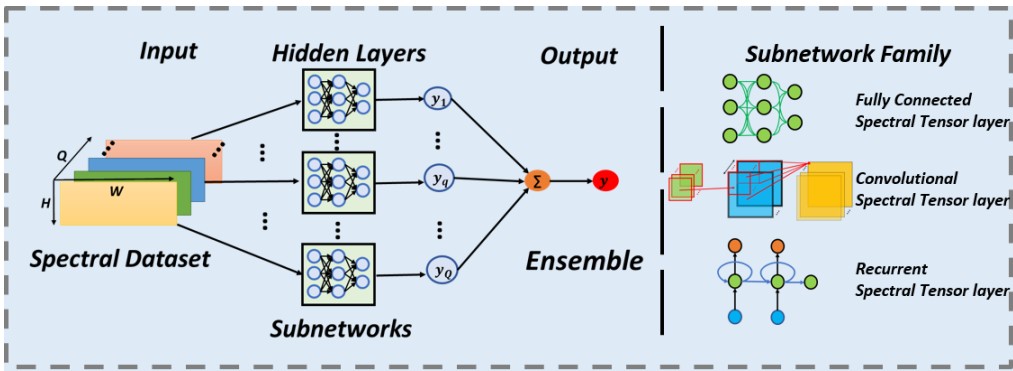

Figure 1: The framework of highly parallel deep ensemble learning.

where the speedup is due to distributing the computation among parallel processors, at the expense of communication overhead.

## 3 Highly Parallel Spectral Tensor Networks

### 3.1 Overview

As shown in Fig.1, a batch of data is firstly pre-processed to the spectral dataset and splitted into $Q$ independent subsets. Then, $Q$ different subnetwork, namely spectral tensor layer, works on $Q$ independent joint spectral dataset to output own result. Last, the final result is the ensemble result from $Q$ subnetworks. Tne subnetwork family includes the fully connected spectral tensor layer, convolutional spectral tensor layer and recurrent spectral tensor layer. The feature of joint data and model parallel makes it suitable for computing on GPUs. The operations in the forward pass, such as t-product further provides potential high parallelism for acceleration on GPUs.

### 3.2 Notations and Basic Tensor Operations

Scalars, vectors, matrices and tensors are denoted by lowercase, boldface lowercase, boldface capital, and calligraphic letters, e.g., $a \in \mathbb{R}$, $\boldsymbol{a} \in \mathbb{R}^n$, $\boldsymbol{A} \in \mathbb{R}^{n_1 \times n_2}$, $\mathcal{A} \in \mathbb{R}^{n_1 \times n_2 \times n_3}$, respectively. We use $\mathcal{A}(:,:,k)$, $\mathcal{A}(:,j,:)$, $\mathcal{A}(i,:,:)$ to denote the frontal, lateral, and horizontal slices.

Given an invertible discrete linear transform $\mathcal{L} : \mathbb{R}^{n_3} \to \mathbb{C}^{n_3}$, let $\mathcal{L}$ and its inverse $\mathcal{L}^{-1}$ be taken along the third-dimension of third-order tensors. That is, for $\mathcal{A} \in \mathbb{R}^{n_1 \times n_2 \times n_3}$, $\widetilde{\mathcal{A}} = \mathcal{L}(\mathcal{A}) \in \mathbb{C}^{n_1 \times n_2 \times n_3}$, with $\widetilde{\mathcal{A}}(i,j,:) = \mathcal{L}(\mathcal{A}(i,j,:))$, $i = 1, ..., n_1, j = 1, ..., n_2$. And for $\widetilde{\mathcal{A}} \in \mathbb{C}^{n_1 \times n_2 \times n_3}$, $\mathcal{A} = \mathcal{L}^{-1}(\widetilde{\mathcal{A}})$, with $\mathcal{A}(i,j,:) = \mathcal{L}^{-1}(\widetilde{\mathcal{A}}(i,j,:))$, $i = 1, ..., n_1, j = 1, ..., n_2$. The general spectral tensor product [41][42][43] is defined as

$$\mathcal{C} = \mathcal{A} \bullet \mathcal{B} = \mathcal{L}^{-1}( \mathcal{L}(\mathcal{A}) \triangle \mathcal{L}(\mathcal{B}) ), \tag{1}$$

where $\triangle$ denotes the frontal-slice wise multiplication, i.e., for $\widetilde{\mathcal{A}} \in \mathbb{C}^{n_1 \times n' \times n_3}, \widetilde{\mathcal{B}} \in \mathbb{C}^{n' \times n_2 \times n_3}$, if $\widetilde{\mathcal{C}} = \widetilde{\mathcal{A}} \triangle \widetilde{\mathcal{B}}$, then $\widetilde{\mathcal{C}}(:,:,k) = \widetilde{\mathcal{A}}(:,:,k) \widetilde{\mathcal{B}}(:,:,k)$, $k = 1, ..., n_3$. The t-product [44] is a special case of (1) where the transform $\mathcal{L}$ is the DFT [45].

For a tensor $\mathcal{A} \in \mathbb{R}^{n_1 \times n_2 \times n_3}$, we define the following operations. $\mathrm{bcirc}(\mathcal{A}) \in \mathbb{R}^{n_1 n_3 \times n_2 n_3}$ organizes its $n_3$ frontal slices into a *block-circulant* matrix

$$\mathrm{bcirc}(\mathcal{A}) = \begin{bmatrix} \mathcal{A}(:,:,1) & \mathcal{A}(:,:,n_3) & \cdots & \mathcal{A}(:,:,2) \\ \mathcal{A}(:,:,2) & \mathcal{A}(:,:,1) & \cdots & \mathcal{A}(:,:,3) \\ \vdots & \vdots & \ddots & \vdots \\ \mathcal{A}(:,:,n_3) & \mathcal{A}(:,:,n_3-1) & \cdots & \mathcal{A}(:,:,1) \end{bmatrix}. \tag{2}$$

Further, $\mathrm{unfold}(\cdot)$ is defined as

$$\mathrm{unfold}(\mathcal{A}) = [\mathcal{A}(:,:,1)^T, \cdots, \mathcal{A}(:,:,n_3)^T]^T \in \mathbb{R}^{n_1 n_3 \times n_2},$$

104 and fold($\cdot$) organizes it back to $\mathcal{A}$, such that

$$\text{fold}(\text{unfold}(\mathcal{A})) = \mathcal{A}. \tag{3}$$

105 Given $\mathcal{A} \in \mathbb{R}^{n_1 \times n' \times n_3}$ and $\mathcal{B} \in \mathbb{R}^{n' \times n_2 \times n_3}$, the t-product [45] can be expressed as follows

$$\mathcal{A} *_t \mathcal{B} = \text{fold}(\text{bcirc}(\mathcal{A}) \cdot \text{unfold}(\mathcal{B})) \in \mathbb{R}^{n_1 \times n_2 \times n_3}. \tag{4}$$

106 The vec($\cdot$) operation maps a matrix in $\mathbb{R}^{n_1 \times n_2}$ into a vector in $\mathbb{R}^{n_1 n_2}$, while vec$^{-1}(\cdot)$ is the inverse
107 mapping.

## 3.3 Fully Connected Spectral Tensor Layer

109 A conventional $N$-layer fully connected network takes $m$ input vectors each of size $\ell'_0 \times 1$ and
110 represents them as a matrix $\boldsymbol{X}^0 \in \mathbb{R}^{\ell'_0 \times m}$. For example, if each input vector represents a color
111 image of size $n \times n \times 3$, then $\ell'_0 = 3n^2$. Each input vector will be classified to one of the $L$ classes.
112 The network parameters at the $j$-th layer include a weight matrix $\boldsymbol{W}^j \in \mathbb{R}^{\ell'_j \times \ell'_{j-1}}$ and an offset
113 $\boldsymbol{B}^j = \underbrace{[\boldsymbol{b}^j, ..., \boldsymbol{b}^j]}_{m} \in \mathbb{R}^{\ell'_j \times m}$, and the forward pass can be represented as [1]

$$\boldsymbol{X}^j = \sigma(\boldsymbol{W}^j \cdot \boldsymbol{X}^{j-1} + \boldsymbol{B}^j), \ \ j = 1, ..., N-1, \tag{5}$$

114 where $\boldsymbol{X}^j \in \mathbb{R}^{\ell'_j \times m}$, and $\sigma(\cdot)$ is an element-wise activation function, e.g., linear, sigmoid, ReLU,
115 and softmax. The last, i.e., $N$-th, layer produces the output $\boldsymbol{Y} \in \mathbb{R}^{L \times m}$ corresponding to the $m$ input
116 vectors, where

$$\boldsymbol{X}^N = \boldsymbol{W}^N \cdot \boldsymbol{X}^{N-1}, \tag{6}$$

$$\boldsymbol{Y} = f(\boldsymbol{X}^N), \tag{7}$$

117 and the output function $f(\boldsymbol{X})$ operates on the columns of $\boldsymbol{X}$, i.e., $f(\boldsymbol{X}(:, s))$ maps $\boldsymbol{X}(:, s)$ to an
118 output score vector $\boldsymbol{Y}(:, s) \in \mathbb{R}^L$ representing the probabilities that the $s$-th input data vector $\boldsymbol{X}^0(:, s)$
119 belongs to different classes. For example $f(\cdot)$ can be a softmax.

120 For our proposed fully connected spectral tensor network, the $m$ input data vectors are organized as a
121 tensor $\mathcal{X}^0 \in \mathbb{R}^{\ell_0 \times m \times Q}$. For the $n \times n \times 3$ color image example, we can set $Q = 3n$ and $\ell_0 = n$,
122 namely each image is a lateral slice of $\mathcal{X}^0$. Using the weight tensor $\mathcal{W}^j \in \mathbb{R}^{\ell_j \times \ell_{j-1} \times Q}$ and offset
123 tensor $\mathcal{B}^j \in \mathbb{R}^{\ell_j \times m \times Q}$, a fully connected tensor layer corresponding to (5) and (6) becomes

$$\mathcal{X}^j = \varrho(\mathcal{W}^j \bullet \mathcal{X}^{j-1} + \mathcal{B}^j), \ j = 1, ..., N-1, \tag{8}$$

$$\mathcal{X}^N = \mathcal{W}^N \bullet \mathcal{X}^{N-1}, \tag{9}$$

124 where $\mathcal{X}^j \in \mathbb{R}^{\ell_j \times m \times Q}$, the spectral tensor product $\bullet$ is given in (1), and the tensor-activation function
125 $\varrho(\cdot)$ under transform $\mathcal{L}$ is defined by applying the conventional element-wise activation function $\sigma(\cdot)$
126 in the spectral domain, i.e.,

$$\varrho(\mathcal{X}) = \mathcal{L}^{-1}(\sigma(\mathcal{L}(\mathcal{X}))). \tag{10}$$

127 A salient feature of the proposed spectral tensor network is the *fully parallel implementation*. Specifi-
128 cally, for $\mathcal{X} \in \mathbb{R}^{\ell \times m \times Q}$, denote $\widetilde{\mathcal{X}} \in \mathbb{C}^{\ell \times m \times Q}$ as the transform of $\mathcal{X}$ along the third dimension, i.e.,
129 $\widetilde{\mathcal{X}}(i, s, :) = \mathcal{L}(\mathcal{X}(i, s, :))$, $i = 1, ..., \ell$, $s = 1, ..., m$. Denote further $\widetilde{\boldsymbol{X}}_q = \widetilde{\mathcal{X}}(:, :, q)$. Then according
130 to (1), (8)-(9) can be split into $Q$ branches of matrix computations

$$\widetilde{\boldsymbol{X}}_q^j = \sigma\left(\widetilde{\boldsymbol{W}}_q^j \cdot \widetilde{\boldsymbol{X}}_q^{j-1} + \widetilde{\boldsymbol{B}}_q^j\right), \tag{11}$$

$$\widetilde{\boldsymbol{X}}_q^N = \widetilde{\boldsymbol{W}}_q^N \cdot \widetilde{\boldsymbol{X}}_q^{N-1}, \ q = 1, ...., Q, \tag{12}$$

131 where $\widetilde{\boldsymbol{X}}_q^j \in \mathbb{C}^{\ell_j \times m}$, $\widetilde{\boldsymbol{B}}_q^j = \underbrace{[\widetilde{\boldsymbol{b}}_q^j, ..., \widetilde{\boldsymbol{b}}_q^j]}_{m} \in \mathbb{C}^{\ell_j \times m}$, and $\widetilde{\boldsymbol{W}}_q^j \in \mathbb{C}^{\ell_j \times \ell_{j-1}}$.

132 We further assume that the weight tensor $\mathcal{W}^j$ in (8)-(9) has low tubal-rank [45][46] such that $\mathcal{W}^j =$
133 $\mathcal{C}^j \bullet \mathcal{D}^j$, where $\mathcal{C}^j \in \mathbb{C}^{\ell_j \times r \times Q}$, $\mathcal{D}^j \in \mathbb{C}^{r \times \ell_{j-1} \times Q}$, and $r \ll \min\{\ell_0, ..., \ell_N\}$. Correspondingly, the
134 weight matrix of each branch has low-rank structure, i.e.,

$$\widetilde{\boldsymbol{W}}_q^j = \widetilde{\boldsymbol{C}}_q^j \cdot \widetilde{\boldsymbol{D}}_q^j, \ q = 1, ..., Q, \tag{13}$$

where $\widetilde{\boldsymbol{C}}_q^j \in \mathbb{C}^{\ell_j \times r}$ and $\widetilde{\boldsymbol{D}}_q^j \in \mathbb{C}^{r \times \ell_{j-1}}$. Then, (11)-(12) become

$$\widetilde{\boldsymbol{Z}}_q^j = \widetilde{\boldsymbol{D}}_q^j \cdot \widetilde{\boldsymbol{X}}_q^{j-1}, \ j = 1, ..., N, \tag{14}$$

$$\widetilde{\boldsymbol{X}}_q^j = \sigma\left(\widetilde{\boldsymbol{C}}_q^j \cdot \widetilde{\boldsymbol{Z}}_q^j + \widetilde{\boldsymbol{B}}_q^j\right), \ j = 1, ..., N-1, \tag{15}$$

$$\widetilde{\boldsymbol{X}}_q^N = \widetilde{\boldsymbol{C}}_q^N \cdot \widetilde{\boldsymbol{Z}}_q^N, \tag{16}$$

where $\widetilde{\boldsymbol{Z}}_q^j \in \mathbb{C}^{r \times m}$, $q = 1, ..., Q$.

Therefore, an $N$-layer fully connected spectral tensor network in (8)-(9) is split into a $2N$-layer network, such that each layer in (11) is now implemented by two sub-layers, namely a linear layer (14) and a nonlinear layer (15), while the $N$-th layer in (12) is implemented by two linear sub-layers, namely (14) and (16). There are multiple parallel matrix multiplications with the same size and along the same dimension in $Q$ subnetworks. Therefore, we employ the batch matrix multiplication using GPUs to accelerate the computations.

Finally, we specify the network output. At each branch $q$, the output function $f(\cdot)$ is applied to the last layer output $\widetilde{\boldsymbol{X}}_q^N$, as in (7), i.e.,

$$\boldsymbol{Y}_q = f(\widetilde{\boldsymbol{X}}_q^N), \ q = 1, ..., Q. \tag{17}$$

Finally the network output is the weighted sum of the outputs of the $Q$ branches, i.e.,

$$\boldsymbol{Y} = \sum_{q=1}^{Q} \omega_q \boldsymbol{Y}_q, \quad \text{s.t. } \omega_q \geq 0, \ \sum_{q=1}^{Q} \omega_q = 1. \tag{18}$$

The loss function can be a cross-entropy function as follows:

$$\text{Loss} = -\sum_{s=1}^{m} \sum_{c=1}^{L} \mathbb{1}(\boldsymbol{y}_s(c) = 1) \cdot \ln(\boldsymbol{Y}(c, s)). \tag{19}$$

Once the spectral tensor network in Fig. 1 is trained, for inference, given a new data sample $\boldsymbol{x} \in \mathbb{R}^{\ell_0 Q}$, we first matricize it into $\boldsymbol{X} \in \mathbb{R}^{\ell_0 \times Q}$ and then take transform along each row to obtain $\widetilde{\boldsymbol{X}}$. We input the $q$-th column of $\widetilde{\boldsymbol{X}}$, i.e., $\widetilde{\boldsymbol{X}}(:, q)$, to the $q$-th sub-network and obtain the output $\boldsymbol{y}_q$, $q = 1, ..., Q$. The final output is then $\boldsymbol{y} = \sum_{q=1}^{Q} \omega_q \boldsymbol{y}_q$.

### 3.4 Convolutional Spectral Tensor Layer

A convolutional neural network [1][47] takes $m$ input images each of dimension $H_0 \times W_0 \times C_0$, i.e., each image has a size $H_0 \times W_0$ and the number of channels is $C_0$, and represents them as a fourth-order tensor $\mathsf{X}^0 \in \mathbb{R}^{H_0 \times W_0 \times C_0 \times m}$. The input to the $j$-th layer is $\mathsf{X}^{j-1} \in \mathbb{R}^{H_{j-1} \times W_{j-1} \times C_{j-1} \times m}$, which is processed by a convolutional kernel $\mathsf{W}^j \in \mathbb{R}^{H_j' \times W_j' \times C_{j-1} \times C_j}$ and an offset $\mathsf{B}^j \in \mathbb{R}^{H_j'' \times W_j'' \times C_j \times m}$, to yield $\mathsf{Y}^j \in \mathbb{R}^{H_j'' \times W_j'' \times C_j \times m}$, with $H_j'' = H_{j-1} - H_j' + 1$, $W_j'' = W_{j-1} - W_j' + 1$, where

$$\mathsf{Y}^j(h, w, c, :) = \sum_{d=1}^{C_{j-1}} \sum_{\ell=0}^{W_j'-1} \sum_{i=0}^{H_j'-1} \mathsf{X}^{j-1}(h+i, w+\ell, d, :) \cdot \mathsf{W}^j(i, \ell, d, c) + \mathsf{B}^j(h, w, c, :), \tag{20}$$

$$h = 1, ..., H_j'', \ w = 1, ..., W_j'', \ c = 1, ..., C_j.$$

To introduce the convolutional spectral tensor network, we represent (20) as a matrix product form that is similar to (5)

$$\boldsymbol{Y}^j = \boldsymbol{W}^j \cdot \boldsymbol{X}^{j-1} + \boldsymbol{B}^j, \tag{21}$$

where $\boldsymbol{Y}^j \in \mathbb{R}^{H_j'' W_j'' C_j \times m}$, $\boldsymbol{W}^j \in \mathbb{R}^{H_j'' W_j'' C_j \times H_{j-1} W_{j-1} C_{j-1}}$, and $\boldsymbol{X}^{j-1} \in \mathbb{R}^{H_{j-1} W_{j-1} C_{j-1} \times m}$ are formed from $\mathsf{Y}^j$, $\mathsf{W}^j$ and $\mathsf{X}^{j-1}$. In particular,

$$\boldsymbol{Y}^j(:, s) = \text{vec}(\text{unfold}(\mathsf{Y}^j(:, :, :, s))),$$
$$\boldsymbol{X}^{j-1}(:, s) = \text{vec}(\text{unfold}(\mathsf{X}^{j-1}(:, :, :, s))), \ s = 1, ..., m. \tag{22}$$

Assume that $C_j = nB$ for $j = 0, ..., N$, then similar to (8)-(9), (21) leads to a convolutional tensor layer

$$\mathcal{Y}^j = \mathcal{W}^j \bullet \mathcal{X}^{j-1} + \mathcal{B}^j, \tag{23}$$

where $\mathcal{Y}^j \in \mathbb{R}^{H_j'' W_j'' n \times m \times B}$, $\mathcal{W}^j \in \mathbb{R}^{H_j'' W_j'' n \times H_{j-1} W_{j-1} n \times B}$, $\mathcal{X}^j \in \mathbb{R}^{H_{j-1} W_{j-1} n \times m \times B}$, and $\mathcal{B}^j \in \mathbb{R}^{H_j'' W_j'' n \times m \times B}$.

Consider the case when $\mathcal{L}$ is DFT, according to (4), (23) can be written as (21) where $\boldsymbol{Y}^j = \text{unfold}(\mathcal{Y}^j)$, $\boldsymbol{X}^{j-1} = \text{unfold}(\mathcal{X}^{j-1})$, $\boldsymbol{B}^j = \text{unfold}(\mathcal{B}^j)$, and $\boldsymbol{W}^j = \text{bcirc}(\mathcal{W}^j) \in \mathbb{R}^{H_j'' W_j'' nB \times H_{j-1} W_{j-1} nB}$ has a block-circulant structure, namely $B \times B$ blocks organized in a circulant form and each block has size $H_j'' W_j'' n \times H_{j-1} W_{j-1} n$. Recall that $\boldsymbol{W}^j$ in (21) is derived from the convolutional kernel $\mathsf{W}^j \in \mathbb{R}^{H_j' \times W_j' \times nB \times nB}$ in (20), following a linear mapping that is consistent with (22). Therefore, the block-circulant structure of $\boldsymbol{W}^j$ in (21) implies a block-circulant structure of each matrix $\mathsf{W}^j(i, \ell, :, :)$ in (20).

Similar to the fully connected case in Section 3.3, the proposed convolutional spectral tensor network also features a *fully parallel implementation*. Specifically, for $\mathcal{X} \in \mathbb{R}^{HWn \times m \times B}$, denote $\widetilde{\mathcal{X}} = \mathcal{L}(\mathcal{X}) \in \mathbb{C}^{HWn \times m \times B}$ as the transform of $\mathcal{X}$ along the third dimension. Denote $\widetilde{\boldsymbol{X}}_b = \widetilde{\mathcal{X}}(:, :, b)$. Then, (23) can be split into $B$ parallel branches of matrix computations as follows

$$\widetilde{\boldsymbol{Y}}_b^j = \widetilde{\boldsymbol{W}}_b^j \cdot \widetilde{\boldsymbol{X}}_b^{j-1} + \widetilde{\boldsymbol{B}}_b^j, \; b = 1, ..., B, \tag{24}$$

where $\widetilde{\boldsymbol{Y}}_b^j \in \mathbb{C}^{H_j'' W_j'' n \times m}$, $\widetilde{\boldsymbol{W}}_b^j \in \mathbb{C}^{H_j'' W_j'' n \times H_{j-1} W_{j-1} n}$, and $\widetilde{\boldsymbol{X}}_b^{j-1} \in \mathbb{C}^{H_{j-1} W_{j-1} n \times m}$. To fully utilize the parallelism of $B$ branches, we employ batch matrix multiplication on GPUs to accelerate (24).

We convert (24) back to the convolutional form in (20), using an inverse mapping of (22) as follows

$$\begin{aligned} \widetilde{\mathsf{Y}}_b^j(:, :, :, s) &= \text{fold}(\text{vec}^{-1}(\widetilde{\boldsymbol{Y}}_b^j(:, s))), \\ \widetilde{\mathsf{X}}_b^{j-1}(:, :, :, s) &= \text{fold}(\text{vec}^{-1}(\widetilde{\boldsymbol{X}}_b^{j-1}(:, s))), \; s = 1, ..., m. \end{aligned} \tag{25}$$

For the $b$-th branch in (24), the input is $\widetilde{\mathsf{X}}_b^{j-1} \in \mathbb{C}^{H_{j-1} \times W_{j-1} \times n \times m}$, the output feature map is $\widetilde{\mathsf{Y}}_b \in \mathbb{C}^{H_j'' \times W_j'' \times n \times m}$, and the kernel weight is $\widetilde{\mathsf{W}}_b^j \in \mathbb{C}^{H_j' \times W_j' \times n \times n}$. Then for $b = 1, ..., B$, (24) can be rewritten as

$$\begin{aligned} \widetilde{\mathsf{Y}}_b^j(h, w, c, :) &= \sum_{d=1}^{n} \sum_{\ell=0}^{W_j'-1} \sum_{i=0}^{H_j'-1} \widetilde{\mathsf{X}}_b^{j-1}(h+i, w+\ell, d, :) \cdot \widetilde{\mathsf{W}}_b^j(i, \ell, d, c) + \widetilde{\mathsf{B}}_b^j(h, w, c, :), \\ h &= 1, ..., H_j'', \; w = 1, ..., W_j'', \; c = 1, ..., n. \end{aligned} \tag{26}$$

Similar to the fully connected tensor networks in Section 3.3, we apply the activation function in the spectral domain as follows

$$\widetilde{\mathsf{Z}}_b^j = \sigma(\widetilde{\mathsf{Y}}_b^j) \in \mathbb{C}^{H_j'' \times W_j'' \times n \times m}. \tag{27}$$

Then, the pooling operation is performed at the $j$-th layer of each branch, resulting in the output $\widetilde{\mathsf{X}}_b^j \in \mathbb{C}^{H_j \times W_j \times n \times m}$.

At the last layer, the output function $f(\cdot)$ is applied to $\widetilde{\mathsf{X}}_b^N$ as in (22).

## 3.5 Recurrent Spectral Tensor Layer

To present the recurrent spectral tensor layer, We take the same case in Section 3.3. Different from the fully connected layer, the network parameters of the recurrent layer at the $j$-th layer include a weight matrix $\boldsymbol{W}_X^j \in \mathbb{R}^{\ell_j' \times \ell_{j-1}'}$, $\boldsymbol{W}_H^j \in \mathbb{R}^{\ell_{j-1}' \times \ell_{j-1}'}$, $\boldsymbol{W}_Y^j \in \mathbb{R}^{\ell_{j-1}' \times \ell_{j-1}'}$ and an offset $\boldsymbol{B}^j = \underbrace{[\boldsymbol{b}^j, ..., \boldsymbol{b}^j]}_{m} \in \mathbb{R}^{\ell_j' \times m}$, and the forward pass can be represented as [1]

$$\begin{aligned} \boldsymbol{H}_t^j &= \sigma_1(\boldsymbol{W}_X^j \cdot \boldsymbol{X}_t^{j-1} + \boldsymbol{W}_H^j \cdot \boldsymbol{H}_t^{j-1} + \boldsymbol{B}^j), \\ \boldsymbol{X}_t^j &= \sigma_2(\boldsymbol{W}_Y^j \cdot \boldsymbol{H}_t^j), \; j = 1, ..., N, t = 1, 2, ..., T, \end{aligned} \tag{28}$$

193    where $\boldsymbol{X}_t^j \in \mathbb{R}^{\ell_j' \times m}$ is the input matrix at the $j$-t time step, and $\boldsymbol{Y}_t^N = \boldsymbol{X}_t^N$.

194    For the introduced recurrent spectral tensor layer, the forward can be computed as

$$\begin{aligned}
\mathcal{H}_t^j &= \varrho_1(\mathcal{W}_X^j \bullet \mathcal{X}_t^{j-1} + \mathcal{W}_H^j \bullet \mathcal{H}_t^{j-1} + \mathcal{B}^j), \\
\mathcal{X}_t^j &= \varrho_2(\mathcal{W}_Y^j \bullet \mathcal{H}_t^j), \;\; j = 1, ..., N, t = 1, 2, ..., T,
\end{aligned} \tag{29}$$

195    where $\mathcal{X}_t^j \in \mathbb{R}^{\ell_j \times m \times Q}$.

196    Furthermore, the implementation of $Q$ branches for computations can be written as

$$\begin{aligned}
\widetilde{H}_{q,t}^j &= \varrho_1(\widetilde{W}_X^j \bullet \widetilde{X}_{q,t}^{j-1} + \widetilde{W}_H^j \bullet \widetilde{H}_{q,t}^{j-1} + \widetilde{B}^j), \\
\widetilde{X}_{q,t}^j &= \varrho_2(\widetilde{W}_Y^j \bullet \widetilde{H}_{q,t}^j), \;\; j = 1, ..., N, t = 1, 2, ..., T,
\end{aligned} \tag{30}$$

197    where $\widetilde{\boldsymbol{X}}_{q,t}^j \in \mathbb{C}^{\ell_j \times m}$ is the input for the $q$-th subnetwork at the $t$-th time step, likewise for $\widetilde{\boldsymbol{H}}_{q,t}^j$.
198    There are $2Q$ parallel matrix multiplications with the same size and along the same dimension, thus
199    we use the batch matrix multiplication on GPUs to accelerate (30).

## 200   4   Performance Evaluation

201    We first describe the experimental settings, then present the results on the MNIST, CIFAR-10 and
202    ImageNet data sets.

### 203   4.1   Data Sets and Performance Metrics

204    We verify the performance of the proposed spectral tensor networks on the following three widely
205    used data sets: 1) MNIST [48] contains grayscale images of handwritten digits. Each image has
206    $28 \times 28$ pixels. The training set has $60,000$ images and the testing set has $10,000$ images. 2)
207    CIFAR-10 [49] contains $60,000$ color images in 10 classes, where each image has size $32 \times 32 \times 3.3$)
208    ImageNet-1K [19]: It contains $12,000,000$ training images and $50,000$ testing images with size of
209    $224 \times 224 \times 3$, labeled with the presence or absence of $1000$ object categories that do not overlap
210    with each other.

211    We are interested in the following performance metrics:1) *Compression ratio*: the ratio of the
212    conventional network size to the spectral tensor network size, which is the also the total reduction in
213    computation due to the reduced number of non-zero network weights; 2) Parallel speedup: the ratio
214    of the training time of a conventional network to that of the spectral tensor network, due to the fully
215    parallel training of all sub-networks; 3) *Convergence*: the loss value versus the training iterations: 4)
216    *Accuracy*: the percentage of correctly estimated labels. Both the training and testing processes are
217    executed on a DGX-2 server [50] that has two $64$ core AMD CPUs, $8$ NVIDIA A100 GPUs and $2$
218    TB of memory. The operating system is Ubuntu $20.04$ with CUDA $10.1$. We use PyTorch [51] to
219    implement neural networks.

220    We summarize the compression ratio, the reduction of computation, and the parallel speedup in
221    Table 2. They are theoretical upper bounds, while their actual values depend on data sets and
222    implementations. For the compression ratio and reduction in computation, each fully connected
223    network / convolutional network has two columns: the right one corresponds to select-the-best
224    weighting, and the left one to other weighting methods.

### 225   4.2   Evaluation of Fully Connected Spectral Tensor Networks

226    For comparison, we consider a conventional fully connected network (FC) [1], the tNN [44], and the
227    fully connected spectral tensor network (FC-tensor) in Section 3. All three methods use the ReLU
228    activation function as $\sigma(\cdot)$ in the hidden layers, the softmax function as the output function $f(\cdot)$ in
229    the last layer, and the cross-entropy loss function in (19). We use $N = 8$ layers in each method and
230    the DCT transform in tNN and the proposed FC-tensor method. The learning rate was set to be $0.01$,
231    the batch size was set to be $64$, and we used the Adam optimizer [52].

232    For the MNIST data set, the conventional FC method has $n = 28, \ell_0' = ... = \ell_7' = 784$, and $L = 10$.
233    Both the tNN method and our FC-tensor method have $n = 28, Q = 28, \ell_0 = ... = \ell_7 = 28$, and
234    $L = 10$. Our FC-tensor method has $r = 8$. For the CIFAR-10 data set, the following parameters are

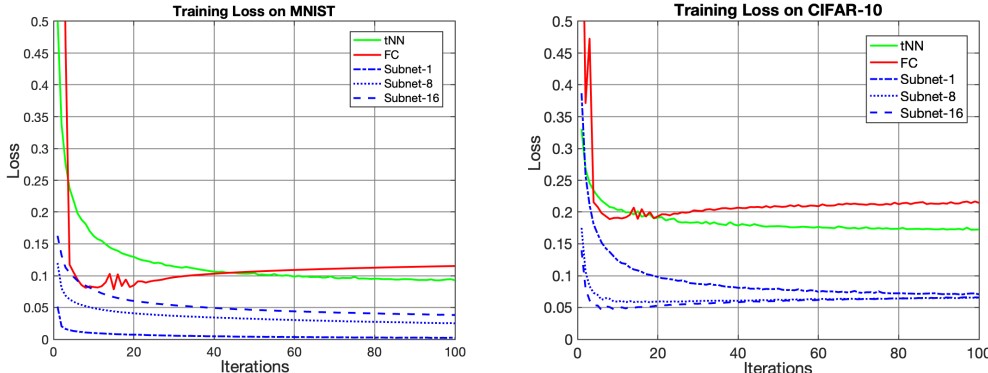

Figure 2: Training loss of fully connected networks on the MNIST data set (left) and CIFAR-10 data set (right).

Table 2: Upper bounds for model compression and parallel speedup. For fully connected networks, $n$ denotes the input size of each sub-network, $r$ is a rank value, and there are $Q$ branches. For convolutional networks, there are $B$ branches.

| - | Fully Connected | Convolutional (1D) |
|---|---|---|
| Compression ratio | $O(nQ/2r), O(nQ^2/2r)$ | $O(B), O(B^2)$ |
| Reduction in computation | $O(nQ/2r), O(nQ^2/2r)$ | $O(B), O(B^2)$ |
| Parallel speedup | $O(Q)$ | $O(B)$ |

different: $n = 32, Q = 32, \ell'_0 = ... = \ell'_7 = 1,024$, and $\ell_0 = ... = \ell_7 = 32$. Therefore, our methods achieve a compression ratio of $49\times$ and $64\times$ for the two data sets, respectively.

The training loss over iterations is shown in Fig. 2, with the left one for the MNIST datas set and the right one for the CIFAR-10 data set. Our scheme converges faster than tNN and FC, while the training process is more stable than FC. The possible reason is that the FC-tensor has much less parameters so that a more stable model can be learned from the same amount of data samples[1]. The loss values of our sub-networks are lower than both tNN and FC.

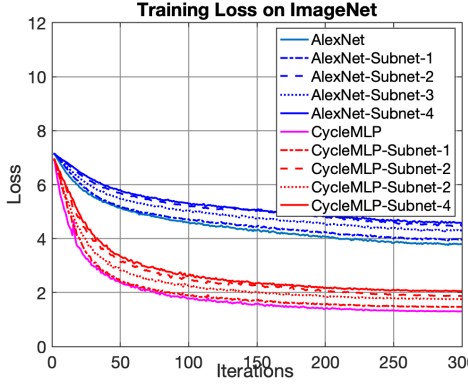

Figure 3: Training loss on ImageNet-1K data set.

Table 3: MNIST and CIAFR-10 data sets.

| Methods | MNIST | CIFAR-10 |
|---|---|---|
| FC [1] | 98.71% | **59.19%** |
| tNN [44] | 97.59% | 44.50% |
| FC-tensor (**average**) | 97.43% | 47.24% |
| FC-tensor (**weighted sum**) | 98.02% | 48.13% |
| FC-tensor (**geometric**) | **99.01%** | 48.33% |

In Table 3, we report accuracy results on both MNIST and CIAFR-10 data sets. Among the four schemes for weighting the sub-networks , the geometric weighting gives the best performance. For the MNIST data set, all three methods achieve a relative high accuracy, i.e., over $97\%$, while our FC-tensor method reaches $98.36\%$. For the CIFAR-10 data set, all three methods achieve a relative low accuracy, i.e., below $60\%$. This is consistent with the known fact that fully connected layers are

---

[1]Note that we use the same number of layers and the same batch size.

Table 3: Results on the ImageNet-1K data set.

| Methods | Accuracy | Size | Training Time |
|---|---|---|---|
| AlexNet [18] | **63.44%** | 244 MB | 40.8 h |
| AlexNet-spectral (**average**) | 61.26% | 130 MB | 31.9 h |
| AlexNet-spectral (**weighted sum**) | 58.01% | 130 MB | 31.9 h |
| AlexNet-spectral (**geometric**) | 62.26% | 130 MB | 31.9 h |
| AlexNet-spectral (**select-the-best**) | 56.45% | 32.5 MB | 31.9 h |
| CycleMLP [53] | **83.23%** | 103 MB | 93.6 h |
| CycleMLP-spectral (**average**) | 78.80% | 76 MB | 60.4 h |
| CycleMLP-spectral (**weighted sum**) | 77.54% | 76 MB | 60.4 h |
| CycleMLP-spectral (**geometric**) | 83.20% | 76 MB | 60.4 h |
| CycleMLP-spectral (**select-the-best**) | 72.45% | 19 MB | 60.4 h |

not enough for the classification task on CIFAR-10. Note that both tNN and FC-tensor achieve lower accuracy than the FC method.

## 4.3 Evaluation on ImageNet Data set

The ImageNet data set [19] is split into $B = 4$ spectral subsets, where each image is organized into a tensor of size $56 \times 56 \times 3 \times 16$ and then processed into a spectrum tensor using DCT transform. Note that the three RGB channels are processed independently.

Our proposed spectral tensor methods have the same structure in Fig. 1, where each branch is replaced by either AlexNet [18] or CycleMLP. We use the DCT transform in our spectral methods. We set the learning rate $0.01$ and the batch size $128$. We follow the standard practice in the community by reporting the top-1 accuracy on the testing set.

For the ImageNet data set, the training loss over training iterations is shown in Fig. 3. Our spectral sub-networks have similar loss curve to their original networks. In Table 3, we report the accuracy, model size, and training time. For the AlextNet structure, our spectral network achieves $1.88\times$ model compression and $1.28\times$ speedup in training time, at the cost of an accuracy drop of $1.18\%$. For the CycleMLP structure, our spectral network achieves $1.36\times$ model compression and $1.55\times$ speedup in training time, at the cost of an accuracy drop of $0.03\%$.

## 5 Conclusions

In this paper, we have proposed a spectral tensor form of deep neural networks that is inherently compressive and allows communication-free parallel/distributed implementations. The data is organized into tensors and a linear transform is applied along certain dimension, resulting in different spectral subsets. The overall network consists of parallel branches of networks, each independently performs training and inference on a spectral data subset. We tested the proposed spectral networks, including fully connected, convolutional, AlexNet, and CycleMLP, on the MNIST, CIFAR-10 and ImageNet data sets, and results show that they can achieve relatively high accuracy with substantial network compression, computation reduction, and parallel speedup, compared with conventional networks. Limited by the pages, We do not provide experiments using the recurrent spectral tensor layer.

For future works, we would like to explore an ensemble-style approach *model soup* [54] that takes average over multiple trained models and achieves state-of-the-art performance on the ImageNet data set.

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
