# OpenReview forum: "Highly Parallel Deep Ensemble Learning"
_NeurIPS.cc/2022/Conference — NeurIPS 2022 Submitted_

### Official Review · Reviewer_JvnU · 2022-07-01

**Rating:** 4
**Confidence:** 2
**Soundness:** 3 good
**Presentation:** 1 poor
**Contribution:** 3 good

**Summary:**

The paper introduces a method to train DNN models using parallel ensemble learning. The key idea is to build networks with parallel branches and update each branch independently using split, independent data sets. The evaluation shows that such a method can lead to reduced computation and model size on MNIST, CIFAR-10, and ImageNet.

**Questions:**

Please describe the relationship between the proposed method with respect to existing parallelism strategies for model training and MoE models.

Why choose AlexNet for the evaluation? It would strengthen the work a lot if it is performed on more standard MLPerf benchmarks, such as ResNet or BERT models.

**Ethics Review Area:**

["I don’t know"]

**Limitations:**

No, there is almost no discussion on the limitations of the proposed method. It would be better if the authors could discuss the rationale behind the chosen workloads and also the generalizability of the proposed method.

**Strengths And Weaknesses:**

Strengths:

- Interesting idea to combine parallel ensemble learning and compression.
- The paper obtains preliminary promising results on training speedups and model sizes.


Weaknesses:

- The proposed method in this paper is highly related to parallel training and compression but appears to miss a lot of related work in both areas. On the training side, there are many existing works on reducing the memory consumption of DNN training, such as [1-3]. On the compression side, the proposed method looks closely related to mixture-of-experts (MoE) where a model has multiple parallel expert branches and each branch is sparsely activated during training, but it misses citing any MoE related papers such as MoE[4], Gshard[5], SwitchTransformer[6]. Overall, parallel ensemble learning is an interesting idea and could lead to something quite impactful. However, given its current status of not positioning it well with many existing works on parallel training and sparse training, it is hard to evaluate the significance and novelty of the proposed method.

There are some mismatched statements. For example, in line 27, the paper says modern DNNs have billions of parameters, but then the examples given are AlexNet, which has only 60 million parameters and is one the earliest network far from representing billion-scale models.

[1] Rajbandari et al. "ZeRO: Memory Optimizations Toward Training Trillion Parameter Models",  https://arxiv.org/pdf/1910.02054.pdf
[2] Shoeybi et al. "Megatron-LM: Training Multi-Billion Parameter Language Models Using Model Parallelism", https://arxiv.org/abs/1909.08053
[3] Huang et al. "GPipe: Efficient Training of Giant Neural Networks using Pipeline Parallelism", https://arxiv.org/abs/1811.06965
[4] Shazeer et al. "Outrageously Large Neural Networks: The Sparsely-Gated Mixture-of-Experts Layer"
[5] Lepikhin et al. "GShard: Scaling Giant Models with Conditional Computation and Automatic Sharding"
[6] Fedus et al. "Switch Transformers: Scaling to Trillion Parameter Models with Simple and Efficient Sparsity"

---

> ### Author Response · Authors · 2022-08-02
> **Response to Reviewer  JvnU**
>
> **Q1:Please describe the relationship between the proposed method with respect to existing parallelism strategies for model training and MoE models.**
>
> A: There are two parallelism strategies, data parallel and model parallel. 1) Data parallel: existing work [1] focused on splitting the datasets into different subsets on time domain, for distributed parallel computation. It reduces the number of samples for MoE models. In our work, we split the datasets into different components on frequency domain. It won’t affect the number of training samples for each model. 2) Model parallel: existing work [1,2,3] focused on partitioning the weight matrix into different blocks and employing the block matrix multiplication to achieve parallel model training.  In our work, we adopts vanilla parallel ensemble training method, which distributes modes of reduced size to different computation nodes.
>
> [1] Rajbandari et al. "ZeRO: Memory Optimizations Toward Training Trillion Parameter Models", https://arxiv.org/pdf/1910.02054.pdf
> [2] Shoeybi et al. "Megatron-LM: Training Multi-Billion Parameter Language Models Using Model Parallelism", https://arxiv.org/abs/1909.08053
> [3] Huang et al. "GPipe: Efficient Training of Giant Neural Networks using Pipeline Parallelism", https://arxiv.org/abs/1811.06965
>
> **Q2:Why choose AlexNet for the evaluation? It would strengthen the work a lot if it is performed on more standard MLPerf benchmarks, such as ResNet or BERT models**.
>
> A: Thanks for your suggestions. Due to the page limit, we did not demonstrate entire experiments. We conduct additional experiments for more standard models, such as ResNet and VGG as follows,
> **Table 1: results of ResNet and VGG on ImageNet.**
> | Methods                   | Acc.(%) | Size(MB) | Time(hours) | Compression Ratio (%) | Speedups |
> |---------------------------|---------|----------|-------------|-----------------------|----------|
> | ResNet                    | 73.60   | 135      | 56.1        | 100                   | 1.0      |
> | ResNet-Spectral (Average) | 69.52   | 66       | 40.8        | 48.8                  | 1.37     |
> | VGG                       | 70.3    | 574      | 89.2        | 100                   | 1.0      |
> | VGG-Spectral (Average)    | 67.8    | 299      | 75.8        | 52.1                  | 1.18     |

---

> > ### Comment · Reviewer_JvnU · 2022-08-08
> > **Response to the rebuttal**
> >
> > Thank you for providing the response. The reviewer appreciates the added new results. However, the concerns are not addressed. In particular,
> >
> > (1) The discussion with respect to existing parallel training and sparse training is still largely inadequate. For example, MoE models also let each expert learn different data, and the exact data distribution is decided by a routing function, which has many proposed strategies, ranging from random, data-aware hashing, and many more [1].
> >
> > (2)The reported results are not sufficient to make conclusions. As can be seen, the proposed method leads to faster training speed and smaller model size, but also much worse accuracy (e.g., >4 points drop in ResNet). In the reviewer's personal experience, if one directly trains a smaller ResNet (the authors did not say which ResNet was used as the baseline given that there were many ResNet variants), the accuracy drop might be smaller than the one from ResNet-Spectral. To be more convincing, it would be better to include results of variants of ResNet with different sizes, such as ResNet-18, ResNet-34, ResNet-50, ResNet-101, and draw Pareto curves to see where ResNet-Spectral fits.
> >
> > [1] BASE Layers: Simplifying Training of Large, Sparse Models, https://arxiv.org/pdf/2103.16716.pdf

---

> > > ### Author Response · Authors · 2022-08-09
> > > **Response to Reviewer JvnU**
> > >
> > > Thanks for your reply! For the first question, in our work, different model learns different spectrals of the data, rather than the different subsets of the data. It is the major difference with conventional MoE models. For the second question, we adopts the ResNet18 as the benchmark. Besides, we really appreciate your suggestions and will add the results in our revision.

---

### Official Review · Reviewer_thRQ · 2022-07-10

**Rating:** 4
**Confidence:** 3
**Soundness:** 3 good
**Presentation:** 2 fair
**Contribution:** 2 fair

**Summary:**

This paper introduces a parallelization of DNN from a tensor perspective. The data is split into their spectral subsets and is passed into parallel subnetworks and finally, the results are based on an ensemble of the subnetworks. Specifically, the author provides detailed spectral counterparts for fully connected layer, convolution layer and recurrent layers. The tensorized structure naturally enables network compression at the same time.

**Questions:**

1. how to make sure that the divided subsets are independent and orthogonal to each other.

2. What is the performance (accuracy, runtime, compression) compared to the sota architectures in the benchmark in the experiment session.

3. Is there any way to demonstrate or analyze the generalization ability of proposed structures compared to original structures?

**Limitations:**

Limitations not discussed.

**Strengths And Weaknesses:**

The overall idea is interesting and novel to some extent. By separating data into spectral subsets, the compression limit is extended as well as the level of parallelization. The experimental supports the claim that applying the technic speeds up the training and the inference.


The overall structure is well-organized. However, there may be space for improvement in conveying the motivation, ideas, and justifications.

In the introduction part, the author mentions that the proposed method has no communication cost. It may be better if the author could support such claims by comparing them with other parallelization ideas which have such costs.

For the main idea, the author mentions the independence of each subset and the orthogonality among each subset. However, how to obtain such subsets and a principal way to select the desired dimension (Q) is not very clear and well-justified.

My main concern with the work, besides the justification, is how significant and general it is.

In the experiment session, the author compares performance and runtime on FC, tNN, AlexNet and CycleMLP, but no state-of-the-art architectures on the benchmarks.

Also there is a lack of runtime comparison on other parallel DNN baselines.

As for the recurrent settings, the attention mechanism also achieves a certain level of parallelization, which may be worth comparing as well.

---

> ### Author Response · Authors · 2022-08-02
> **Response to Reviewer thRQ**
>
> **Q1:how to make sure that the divided subsets are independent and orthogonal to each other. **
>
> A: On the data processing stage, we put DCT transformation on the input dataset, for which the transformed dataset is represented by a series of orthogonal basis on frequency domain. Then, we manually split the dataset into different sub datasets which are then sent into different networks, which guarantees the inputs for each network are different and orthogonal to each other.
>
> **Q2:What is the performance (accuracy, runtime, compression) compared to the sota architectures in the benchmark in the experiment session.**
>
> A: CycleMLP is one of the sota architecture for ImageNet classificaiton and has been reported in our experiment session.
>
> **Q3:Is there any way to demonstrate or analyze the generalization ability of proposed structures compared to original structures?**
>
> A: Thanks for your review. In our opinion, the performance on test dataset can be regarded as the generalization ability. Thus, the proposed structure has a similar performance compared with original structures.

---

> > ### Comment · Reviewer_thRQ · 2022-08-08
> > **Response to the authors**
> >
> > I would like to appreciate the reply and the effort for further explanation. However, there are still some concerns about the significance of the proposed methodology. The provided experimental results show that the advantage of the parallel structure (speed up) is sometimes traded by model performance. There are other forms of parallel/compression algorithms that obtain faster runtime with similar or less performance drop. The proposed method does not indicate why it stands out from its counterparts. And it would be better if the authors provide more justification or insights for the design. Overall, the proposed architecture shade light on some interesting angles of parallelization/data-side ensemble, but could be in better shape and may not be ready for publication.

---

### Official Review · Reviewer_evpK · 2022-07-12

**Rating:** 4
**Confidence:** 4
**Soundness:** 2 fair
**Presentation:** 1 poor
**Contribution:** 2 fair

**Summary:**

The authors proposed a spectral tensor decomposition of a DNN into independent, parallel and computationally cheaper subnets that process corresponding spectrally decomposed data, resulting in speedup in training; ensembling of these subnets yielded reasonable generalization performance.


**Questions:**

- Though a recurrent spectral tensor layer was derived, it was not demonstrated with experiments.
- The example DNNs experimented are not representative or practically relevant.
- Do spectral tensor transformers exist?
- It seems decomposition of data with orthogonal bases is not unique.  Nor is the number of components and the grouping thereof.  How does one choose optimal or reasonable decomposition?
- With a certain data decomposition, task-relevant information in different components naturally vary.  Corresponding subnet experts might need to have different capacities to realize maximal efficiency?  Or how does one equalize component importance, or do spectral pruning?
- Fig. 2, why was the FC training numerically unstable, with a non-monotonic training loss curve?



**Limitations:**

See above.

**Strengths And Weaknesses:**

[+] Novel idea

[-] Inadequate supporting experiments

---

> ### Author Response · Authors · 2022-08-02
> **Response to Reviewer evpK**
>
> **Q1:Though a recurrent spectral tensor layer was derived, it was not demonstrated with experiments.**
>
> A: Due to the page limit, we did not demonstrate experiments to present the recurrent spectral tensor layer. We additionally conduct experiments on MNIST using recurrent spectral tensor layer. Specifically, We construct a neural network consisting of a LSTM unit with a 64 neuron hidden layer and a fully connected layer for classification. We use the SGD optimizer and set the learning rate as 0.01, the batch size as 64 and the training epochs as 10. The results are listed as follows,
>
> **Table 1: Results of RNN on MNIST.**
> | Method                              | Acc.(%) |
> |-------------------------------------|---------|
> | RNN+FC                              | 94.6    |
> | RNN-tensor+FC-tensor (average)      | 93.7    |
> | RNN-tensor+FC-tensor (weighted sum) | 94.1    |
> | RNN-tensor+FC-tensor (geometric)    | 95.9    |
>
>
>
> **Q2: The example DNNs experimented are not representative or practically relevant.**
>
> A: In our experiments, the reason why we use a simple DNN is to verify the correctness and effectiveness of our fully connected spectral tensor layer.
>
> **Q3:Do spectral tensor transformers exist?**
>
> A:  No. We haven't implement a spectral tensor transformer.
>
> **Q4: It seems decomposition of data with orthogonal bases is not unique. Nor is the number of components and the grouping thereof. How does one choose optimal or reasonable decomposition?**
>
> A: There are many methods for the data split and we select the best method by the performance, i.e., the classification accuracy.
>
> **Q5: With a certain data decomposition, task-relevant information in different components naturally vary. Corresponding subnet experts might need to have different capacities to realize maximal efficiency? Or how does one equalize component importance, or do spectral pruning?**
>
> A: We have the weighted sum on the ensemble process, e.g., weighted sum and geometric.
>
> **Q6:Fig. 2, why was the FC training numerically unstable, with a non-monotonic training loss curve?**
>
> A: It happens on the begining stage of the training process of DNN and it is normal for the training. When the loss converges, there is no numerical unstability.

---

### Official Review · Reviewer_sYxS · 2022-07-12

**Rating:** 6
**Confidence:** 4
**Soundness:** 3 good
**Presentation:** 3 good
**Contribution:** 4 excellent

**Summary:**

This work proposes a tensorial representation of data, and linear transformation and split into multiple spectral subsets. An independent neural network is trained on each of these subsets at training and combined through an ensembling approach. This reduces the computational complexity due to smaller data size per network and the parallel training of each subnetwork.

**Questions:**

* In related works, it would be helpful if the proposed approach is contrasted against other ensembling techniques that learn sparse subnetworks [1, 2]. Also, discussion on other data subsetting approaches []  will provide a good context.

* For the fully connected spectral tensor layer, is it possible to set Q < 3n? if so, has the sensitivity to this been evaluated? Similarly, is the effect of the choice of B for the convolutional spectral tensor layer evaluated?

* It is mentioned that the performance metrics of Compression ratio, Parallel Speedup, Convergence, and Accuracy are considered, but these metrics are not explicitly shown in table 3 and 4
* How are the neural network hyperparmeters chosen for MNIST and CIFAR-10 FC-tensor method?
* The performance of tNN and FC-tensor seems to be poor in CIFAR-10 compared to a vanilla FC, why is this?
* Why not use a convolutional spectral tensor layer for CIFAR10.
* Is the O(Q) and O(B) speedup seen during training?
* A discussion on the limitations of the approach will be helpful. Can this be applied for residual networks?
* Minor typos in multiple places
   * Strucred linear layers (pg 1)
   * CIAFR-10 used multiple times (pg 8)



**Limitations:**

the limitations and potential negative societal impact of this work is not provided

**Strengths And Weaknesses:**


**Strengths**

* Able to train an ensemble of subnetworks on a subset of data in a highly parallel fashion is an efficient way to learn generalizable and efficient neural network model.
* This can lead to a paradigm shift in the way models are learned and can enable the scaling of approach to larger data and network sizes

**Weaknesses**
The (novel) contribution of the paper and contrast with literature is not clear. The tensor layer idea seems to be already proposed in the literature. If the parallel implementation and ensembling the network outputs is the contribution, it should be made clear

---

> ### Author Response · Authors · 2022-08-02
> **Response to Reviewer sYxS Part 1**
>
> **Q1: In related works, it would be helpful if the proposed approach is contrasted against other ensembling techniques that learn sparse subnetworks [1, 2]. Also, discussion on other data subsetting approaches [] will provide a good context.**
>
> A: Thanks for your suggestion. We agree that it would help strengthen the paper quality if adding some discussion on the contrast with other ensembling techniques.
> As you mentioned, the ensembling techniques that learn sparse subnetworks, namely dynamic sparse training ensemble (DST Ensemble) [1] connect sparse training with deep ensemble methods to reduce parameters and FLOPs, thus improving the training efficiency. Also, some work [2] employs data subsetting approaches to reduce computation and communication overhead.
> Our work adopts a different ensembling method to achieve data and model parallelism. First, we split the dataset into different components with reduced size in frequency domain, while previous work focuses on splitting the datasets into different subsets. Second, with the size of inputs reduced, the size of models can be reduced, including the depth and the width of each neural layer, which helps reduce the computation cost for each model. Third, there is no specific assumption about the model, including the sparsity and architecture. Thus, it is easy to compatible our method with other ensembling techniques to achieve better performance.
>
> [1] Liu, Shiwei ; Chen, Tianlong ; Atashgahi, Zahra ; Chen, Xiaohan ; Sokar, Ghada ; Mocanu, Elena ; Pechenizkiy, Mykola ; Wang, Zhangyang ; Mocanu, Decebal Constantin. / Deep Ensembling with No Overhead for either Training or Testing : The All-Round Blessings of Dynamic Sparsity. The Tenth International Conference on Learning Representations, ICLR 2022. OpenReview, 2022.
> [2] Bibikar, Sameer & Vikalo, Haris & Wang, Zhangyang & Chen, Xiaohan. (2022). Federated Dynamic Sparse Training: Computing Less, Communicating Less, Yet Learning Better. Proceedings of the AAAI Conference on Artificial Intelligence. 36. 6080-6088. 10.1609/aaai.v36i6.20555.
>
> **Q2: For the fully connected spectral tensor layer, is it possible to set Q < 3n? if so, has the sensitivity to this been evaluated? Similarly, is the effect of the choice of B for the convolutional spectral tensor layer evaluated?**
>
> A: It is possible to set Q <3n while it achieves the best performance when Q=3n. In fact, we have tried many strategies for the split and transformation of the data. However, due to the page limit, we did not demonstrate clearly. For example, for the classification of MNIST dataset, the results of different number of subnetworks are as follows.
> **Table 1: Results of different number of subnetworks.**
> |Number of Subnetworks|Acc.(%)|
> |-------|----|
> |7	|97.78|
> |14	|97.79|
> |16	|97.86|

---

> > ### Author Response · Authors · 2022-08-02
> > **Response to Reviewer sYxS Part 2**
> >
> > **Q3:It is mentioned that the performance metrics of Compression ratio, Parallel Speedup, Convergence, and Accuracy are considered, but these metrics are not explicitly shown in Table 3 and 4**
> > A: Thanks for your suggestions! We agree that it is not clearly shown for the comparison of given metrics on Table 3 and Table 4, especially compression ratio and speedups. For Table 4, we supplements the metrics as follows.
> > **Table 2: Results on ImageNet.**
> > | Methods                             | Acc.(%) | Size(MB) | Time(hours) | Compression Ratio (%) | Speedups |
> > |-------------------------------------|---------|----------|-------------|-----------------------|----------|
> > | AlexNet                             | 63.44   | 244      | 40.8        | 100                   | 1.0      |
> > | AlexNet-Spectral (Average)          | 61.26   | 130      | 31.9        | 53.5                  | 1.28     |
> > | AlexNet-Spectral (Weighted-sum)     | 58.01   | 130      | 31.9        | 53.5                  | 1.28     |
> > | AlexNet-Spectral (Geometric)        | 62.26   | 130      | 31.9        | 53.5                  | 1.28     |
> > | AlexNet-Spectral (Select-the-best)  | 56.45   | 32.5     | 31.9        | 13.3                  | 1.28     |
> > | CycleMLP                            | 82.23   | 103      | 93.6        | 100                   | 1.55     |
> > | CycleMLP-Spectral (Average)         | 78.80   | 76       | 60.4        | 73.8                  | 1.55     |
> > | CycleMLP-Spectral (Weighted-sum)    | 77.54   | 76       | 60.4        | 73.8                  | 1.55     |
> > | CycleMLP-Spectral (Geometric)       | 83.20   | 76       | 60.4        | 73.8                  | 1.55     |
> > | CycleMLP-Spectral (Select-the-best) | 72.45   | 19       | 60.4        | 18.4                  | 1.55     |
> >
> >
> >
> > **Q4: How are the neural network hyperparmeters chosen for MNIST and CIFAR-10 FC-tensor method?**
> >
> > A: As commonly used techniques in other work [1][2], we use the GridSearch Method for the hyper-parameters for training networks on MNSIT and CIFAR-10. Specifically, we first manually split the dataset into train, val and test sub datasets. Then, we use different set of hyper-parameters to train our model on the training dataset and evaluate the performance on the eval dataset. We select the set of hyper-parameters which performs well on val dataset as the final hyper-parameters for training.
> >
> > [1] Bergstra, James, and Yoshua Bengio. "Random search for hyper-parameter optimization." Journal of machine learning research 13.2 (2012).
> > [2] Xiao, Li, Zeliang Zhang, and Yijie Peng. "Noise Optimization for Artificial Neural Networks." arXiv preprint arXiv:2102.04450 (2021).
> >
> >
> > **Q5: The performance of tNN and FC-tensor seems to be poor in CIFAR-10 compared to a vanilla FC, why is this? **
> >
> > A: The number of parameters in tNN and FC-tensor has been largely reduced compared with conventional fully connected layers, namely around 60% compression ratio, which results in a drop of performance on CIFAR10, namely 10% classification accuracy, which is more complicated than MNIST dataset. For generally used models on ImageNet, Over-parametrization is a common issue which helps models behave well even with a reduction of parameters as shown in our ImageNet experiment.
> >
> > **Q6:Why not use a convolutional spectral tensor layer for CIFAR10.**
> >
> > A: In fact, we have made experiements on CIFAR10. However, due to the page limit, we did not present all the results. Besides, it is more suitable to present the performance of convolutional spectral tensor layer on ImageNet, while MNIST and CIFAR10 are demo datasets.  The results of convolutional spectral tensor layer are as follows.
> > **Table 3: Results of CNN on CIFAR10.**
> > |Modles|Acc.(%)|
> > |-------|----|
> > |CNN	|90.66|
> > |CNN-Spectral	|91.37|

---

> > > ### Author Response · Authors · 2022-08-02
> > > **Response to Reviewer sYxS Part 3**
> > >
> > > **Q7:Is the O(Q) and O(B) speedup seen during training? (potential, ongoing work)**
> > >
> > > A: There remains great  potential for GPU acceleration. It is our onging work which exploits the parallelsim of spectral tensor layer to achieve high performance tensor neural network training.
> > >
> > > **Q8:A discussion on the limitations of the approach will be helpful. Can this be applied for residual networks?**
> > >
> > > A: Thanks for your suggestion. We have experiments on residual networks. Due to the page limit, it is not shown in this version. The result for residual network on ImageNet is as follows,
> > > **Table 3: Results of ResNet on CIFAR10.**
> > > | Methods                   | Acc.(%) | Size(MB) | Time(hours) | Compression Ratio (%) | Speedups |
> > > |---------------------------|---------|----------|-------------|-----------------------|----------|
> > > | ResNet                    | 73.60   | 135      | 56.1        | 100                   | 1.0      |
> > > | ResNet-Spectral (Average) | 69.52   | 66       | 40.8        | 48.8                  | 1.37     |

---

### Author Response · Authors · 2022-08-06
**General response to all reviewers**

We thank the reviewers for their time and extensive feedback. Their comments and suggestions will help us to improve the quality of the manuscript. We have detailed response to each review's comment individually below. However, we also highlight here the main contribution of the paper.

## Contribution of paper
The main purpose of the work presented here is to introduce a highly parallel  deep ensemble learning, which leads to highly compact and  parallel deep neural networks. The key features of our work falls on two aspects, including the highly parallelism and the ensemble learning. For the highly parallelism, different from conventional methods, which utilizes the block matrix multiplication to put one model onto multi-nodes for parallel computation, our work put multiple models of reduced size onto multi-nodes.It benefits from the inputs of reduced size and contributes  to make full use of the GPU power to enable high performance parallel computation. For the ensemble leanring,  different from the conventional MoE, which mainly focus on the ensemble of multiple weak models, our work utilizes the spectral transformation and data split to make different model learn different sub-components independently, wchich potentially improve the generalization and robustness.

---

### Meta-Review · Area_Chair_bEox · 2022-08-24

**Recommendation:** Reject
**Confidence:** Certain

**Metareview:**

Reviewers generally agreed that the parallelization idea described in the manuscript is interesting, but for the most part felt that it is not well positioned within existing literature (making it difficult to gauge the exact novelty), and that empirical results are not sufficiently convincing.  I encourage the authors to address these two critiques and submit their work to a future venue.

**Award:**

No

---

### Decision · Program_Chairs · 2022-09-14

Reject